# Essential Oils, Phytoncides, Aromachology, and Aromatherapy—A Review

Subramanian Thangaleela [1], Bhagavathi Sundaram Sivamaruthi [2,*], Periyanaina Kesika [2], Muruganantham Bharathi [1], Wipada Kunaviktikul [3], Areewan Klunklin [3], Chatnithit Chanthapoon [4] and Chaiyavat Chaiyasut [1,*]

[1] Innovation Center for Holistic Health, Nutraceuticals, and Cosmeceuticals, Faculty of Pharmacy, Chiang Mai University, Chiang Mai 50200, Thailand; drthangaleela@gmail.com (S.T.); bharathi.m@cmu.ac.th (M.B.)
[2] Office of Research Administration, Chiang Mai University, Chiang Mai 50200, Thailand; kesika.p@cmu.ac.th
[3] School of Nursing, Panyapiwat Institute of Management, 85/1 Moo 2 Chaengwattana Road, Bang-Talat, Pakkred, Nonthaburi 11120, Thailand; wipadakun@pim.ac.th (W.K.); areewanklu@pim.ac.th (A.K.)
[4] Research and Development Unit, Chernara Co., Ltd., Chiang Rai 57170, Thailand; chatnithitc@gmail.com
* Correspondence: sivamaruthi.b@cmu.ac.th (B.S.S.); chaiyavat@gmail.com (C.C.)

**Abstract:** Chemical compounds from plants have been used as a medicinal source for various diseases. Aromachology is a unique field that studies the olfactory effects after inhaling aromatic compounds. Aromatherapy is a complementary treatment methodology involving the use of essential oils containing phytoncides and other volatile organic compounds for various physical and mental illnesses. Phytoncides possess an inherent medicinal property. Their health benefits range from treating stress, immunosuppression, blood pressure, respiratory diseases, anxiety, and pain to anti-microbial, anti-larvicidal, anti-septic, anti-cancer effects, etc. Recent advancements in aromatherapy include forest bathing or forest therapy. The inhalation of phytoncide-rich forest air has been proven to reduce stress-induced immunosuppression, normalize immune function and neuroendocrine hormone levels, and, thus, restore physiological and psychological health. The intricate mechanisms related to how aroma converts into olfactory signals and how the olfactory signals relieve physical and mental illness still pose enormous questions and are the subject of ongoing research. Aromatherapy using the aroma of essential oils/phytoncides could be more innovative and attractive to patients. Moreover, with fewer side effects, this field might be recognized as a new field of complementary medicine in alleviating some forms of physical and mental distress. Essential oils are important assets in aromatherapy, cosmetics, and food preservatives. The use of essential oils as an aromatherapeutic agent is widespread. Detailed reports on the effects of EOs in aromatherapy and their pharmacological effects are required to uncover its complete biological mechanism. This review is about the evolution of research related to phytoncides containing EOs in treating various ailments and provides comprehensive details from complementary medicine.

**Keywords:** aromachology; aromatherapy; essential oils; phytoncides; forest bathing

## 1. Introduction

*Aromachology and Aromatherapy*

The emerging study of aromachology first began in the late 20th century, initiated by a scientist from Japan, Shizuo Torii. Torii studied the association between aroma and emotion and found that the fragrance of lavender and chamomile enhances relaxation. The term aromachology was first coined by the Sense of Smell Institute in 1982. Aroma has the extraordinary power to amend our physical, mental, and emotional state of well-being. More than 3500 years ago, the fragrance was used in human life rituals by ancient Egyptians. Our modern frenetic, chaotic lifestyle has brought back the use of aromas to obtain better physical and mental health [1]. Aromachology is the study of the interrelationship between

aroma and psychology. In detail, it is the effect of aroma on the brain, the subsequent subtle neurological and behavioral changes, and the intertwined psychological variations. It has been an emerging and fascinating subject in scientific society for the past few years. The influence of aroma on humans is a new scientific discipline; hence, the need for independent research papers regarding aromachology is clear [2–4].

Aromatherapy is an ancient concept used by the Chinese, Egyptians, and Romans in incense, their baths, and embalming the dead. The word aromatherapy was first coined by the French chemist Rene-Maurice Gattefosse in the 1920s. His first discovery of the healing nature of lavender essential oil was through serendipity when he accidentally soaked his burnt hand in pure lavender oil and found that his hand was rapidly healing. His exploration of essential oils and his experiments in their healing nature was initiated then. In addition to their healing nature, these aroma oils can influence mood, behavior, and wellness [2]. Scientifically defined, aromatherapy is a complementary treatment methodology [5] using essential oils containing phytoncides [6] as a tool for therapeutics [2,3].

Essential oils (EOs) are the secondary metabolites [7] of aromatic plants, representing a complex mixture of volatile organic compounds (VOCs) [4]. The plants hold these oils throughout their thallus structures, such as reservoirs, glandular hairs, special cells, and intracellular spaces. Plants are also protected from pathogenic encounters and temperature fluctuations with the help of these essential oils [8]. EOs are a concoction of chemical groups such as alcohols, ketones, esters, ethers, aldehydes, oxides, phenols, saturated and unsaturated hydrocarbons, and terpenes [9] that can be extracted from different regions of plants, such as the bark of plants, flower petals, stems, leaves, roots, and distillation from resins [10]. EOs' extraction can be carried out by conventional methods such as steam distillation, hydro-distillation, hydro-diffusion, and solvent extraction, and by advanced methods such as supercritical fluid extraction, subcritical extraction, solvent-free microwave-assisted extraction [11], and also by the physical crushing of the outermost waxy layer where oil glands are situated [12]. Several studies have evaluated the therapeutic effects of EOs [13] as additives and packaging materials in the food industry [14] and as air quality enhancers in indoor environments [15]. Some of the well-known aromatic essences are lavender oil, rosemary oil, jasmine oil, and peppermint oil, which have been found to improve cognitive functions, memory retention, pain relief, and mood enhancement and play a role in enhancing physical and psychological conditions affected by stress [16–18]. EOs may be administered through massage, inhalation, or direct application over the skin or internally [19,20]. However, although inhaling aromas in treating ailments or stress is approved as aromatherapy, its effectiveness is still in question.

Though little evidence supports its efficacy and uncertainty because of the scarcity of studies and insufficient understanding related to aromatherapy [2,21], studies have shown that inhaling aroma at night elicits feelings of sensuality and relaxation, happiness, or exhilaration [2,4,22]. Phytoncides are volatile organic substances extracted from plants that possess antimicrobial activities and help in enhancing immune functions through NK cell activity and anti-inflammatory properties. 'Phyton' refers to plants, and 'cide' refers to killing in Greek, thus highlighting the anti-microbial activities [23]. However, the lack of convincing studies is also accountable for our insufficient understanding. Hence, as part of the emerging investigations focused on aromachology and aromatherapy, the present review is designed to assess the effects of aromatherapy on physiological functions.

## 2. Chemistry of Essential Oils

Essential oils are a complex mixture of volatile lipophilic aromatic and aliphatic compounds [24]. They are colorless liquids, soluble in alcohol, ether, and fixed oils but insoluble in water, and they possess a particular odor [25]. EOs combine more than 100 single substances, including terpenoids, phenylpropanoids, and short-chain aliphatic hydrocarbon derivatives. The structure of EOs can contain allylic, bi-, tricyclic, mono-, and sesquiterpenoids of different functional groups, such as hydrocarbons, ketones, and alcohols as well as oxides, aldehydes, phenols, or esters [9]. The loss or absence of any

one of the components in the EOs can change their aroma. Oxygenated EOs are more fragrant than the EOs containing monoterpene hydrocarbons [26]. EOs with C5 units are termed monoterpenes. Monoterpenes are the major component of EOs, responsible for aroma and flavor, and they act as important ingredients in the agricultural, pharmaceutical, cosmetic, and food industries [27]. They include different types, such as nerol, linalool, citronellol, citronellal, and citral [28]. Sesquiterpenes usually contain 15 carbons in their skeleton structures with different functional groups such as hydrocarbons, aldehydes, and alcohols [29]. Sesquiterpenes are responsible for anti-microbial, anti-fungal, anti-tumor, and anti-inflammatory activities [30]. Many factors influence the chemistry of EOs, including the plant organ, genetic factors, geographical variations, environmental conditions, type of species the plant belongs to, mode of production, storage conditions, etc. [31–33]. Many components, such as citral, myrcene, ocimene, menthol, D-limolene, α-pinene, α-thujone, β-thujone, farnesol, α-bisabolol, humelene, etc., are present in EOs and have been proven to be responsible for the effects of EOs. Any small change in these components may change the aroma properties of EOs. The chemical structure of EO components is given in Table 1.

**Table 1.** Structure of the representative volatile aromatic terpenes. Structures have been adapted from the PubChem database (https://pubchem.ncbi.nlm.nih.gov/) (accessed on 9 April 2022).

| S. No | Type of Terpenoids | Component | 2D Structure | 3D Conformer |
|---|---|---|---|---|
| 1 | Acylic Monoterpenes | Citral <br> Pubchem ID 638011 <br> [1] https://pubchem.ncbi.nlm.nih.gov/compound/Citral#section=2D-Structure <br> [2] https://pubchem.ncbi.nlm.nih.gov/compound/Citral#section=3D-Conformer | | |
| 2 | Acylic Monoterpenes | Myrcene <br> Pubchem ID 31253 <br> [1] https://pubchem.ncbi.nlm.nih.gov/compound/myrcene#section=2D-Structure <br> [2] https://pubchem.ncbi.nlm.nih.gov/compound/myrcene#section=3D-Conformer | | |
| 3 | Acylic Monoterpenes | Ocimene <br> Pubchem ID 5281553 <br> [1] https://pubchem.ncbi.nlm.nih.gov/compound/E_-beta-Ocimene#section=2D-Structure <br> [2] https://pubchem.ncbi.nlm.nih.gov/compound/E_-beta-Ocimene#section=3D-Conformer | | |
| 4 | Monocyclic monoterpenes | Menthol <br> Pubchem ID 1254 <br> [1] https://pubchem.ncbi.nlm.nih.gov/compound/Menthol#section=2D-Structure <br> [2] https://pubchem.ncbi.nlm.nih.gov/compound/Menthol#section=3D-Conformer | | |
| 5 | Monocyclic monoterpenes | D-Limolene <br> Pubchem ID 440917 <br> [1] https://pubchem.ncbi.nlm.nih.gov/compound/D-Limonene#section=2D-Structure <br> [2] https://pubchem.ncbi.nlm.nih.gov/compound/D-Limonene#section=3D-Conformer | | |

**Table 1.** *Cont.*

| S. No | Type of Terpenoids | Component | 2D Structure | 3D Conformer |
|---|---|---|---|---|
| 6 | Bicyclic monoterpenes | α-Pinene Pubchem ID 6654 [1] https://pubchem.ncbi.nlm.nih.gov/ compound/alpha-Pinene#section=2D-Structure [2] https://pubchem.ncbi.nlm.nih.gov/ compound/alpha-Pinene#section=3D-Conformer | | |
| 7 | Bicyclic monoterpenes | α-Thujone Pubchem ID 261491 [1] https://pubchem.ncbi.nlm.nih.gov/ compound/alpha-Thujone#section=2D-Structure [2] https://pubchem.ncbi.nlm.nih.gov/ compound/alpha-Thujone#section=3D-Conformer | | |
| 8 | Bicyclic monoterpenes | β-Thujone Pubchem ID 91456 [1] https://pubchem.ncbi.nlm.nih.gov/ compound/beta-Thujone#section=2D-Structure [2] https://pubchem.ncbi.nlm.nih.gov/ compound/beta-Thujone#section=3D-Conformer | | |
| 9 | Acyclic sequiterpenes | Farnesol Pubchem ID 445070 [1] https://pubchem.ncbi.nlm.nih.gov/ compound/Farnesol#section=2D-Structure [2] https://pubchem.ncbi.nlm.nih.gov/ compound/Farnesol#section=3D-Conformer | | |
| 10 | Acyclic sequiterpenes | Nerolidol Pubchem ID 5284507 [1] https://pubchem.ncbi.nlm.nih.gov/ compound/Nerolidol#section=2D-Structure [2] https://pubchem.ncbi.nlm.nih.gov/ compound/Nerolidol#section=3D-Conformer | | |
| 11 | Monocyclic sequiterpenes | α-Bisabolol Pubchem ID 10586 [1] https://pubchem.ncbi.nlm.nih.gov/ compound/alpha-Bisabolol#section= 2D-Structure [2] https://pubchem.ncbi.nlm.nih.gov/ compound/alpha-Bisabolol#section= 3D-Conformer | | |
| 12 | Monocyclic sequiterpenes | Humulene Pubchem ID 5281520 [1] https://pubchem.ncbi.nlm.nih.gov/ compound/Humulene#section=2D-Structure [2] https://pubchem.ncbi.nlm.nih.gov/ compound/Humulene#section=3D-Conformer | | |

**Table 1.** *Cont.*

| S. No | Type of Terpenoids | Component | 2D Structure | 3D Conformer |
|---|---|---|---|---|
| 13 | Bicyclic sequiterpenes | Nootkatone Pubchem ID 1268142 [1] https://pubchem.ncbi.nlm.nih.gov/compound/Nootkatone#section=2D-Structure [2] https://pubchem.ncbi.nlm.nih.gov/compound/Nootkatone#section=3D-Conformer | | |
| 14 | Bicyclic sequiterpenes | Chamazulene Pubchem ID 10719 [1] https://pubchem.ncbi.nlm.nih.gov/compound/Chamazulene#section=2D-Structure [2] https://pubchem.ncbi.nlm.nih.gov/compound/Chamazulene#section=3D-Conformer | | |
| 15 | Tricyclic Sequiterpenes | Thujopsene Pubchem ID 442402 [1] https://pubchem.ncbi.nlm.nih.gov/compound/Thujopsene#section=2D-Structure [2] https://pubchem.ncbi.nlm.nih.gov/compound/Thujopsene#section=3D-Conformer | | |
| 16 | Tricyclic Sequiterpenes | Patchouli alcohol Pubhem ID 10955174 [1] https://pubchem.ncbi.nlm.nih.gov/compound/Patchoulol#section=2D-Structure [2] https://pubchem.ncbi.nlm.nih.gov/compound/Patchoulol#section=3D-Conformer | | |

[1] Source link for 2D structure of the compounds; [2] Source link for 3D structure of the compounds.

## 3. Methodology

The keywords 'aromachology', 'aromatherapy', and 'aromatherapy and phytoncides' were used to search the scientific databases PubMed, Google Scholar, Medline, and PsycINFO. The search for aromatherapy returned 3296 results consisting of both reviews and research articles spanning the choice of aromatherapy for dementia, depression, stress, cognitive dysfunction, anxiety, cardiovascular diseases, etc.

The search for aromatherapy and phytoncides yielded less than a hundred articles, highlighting forest bathing, phytochemicals, essential oils, and volatile organic compounds, and their effects on the immune system and against certain viral infections. The total number of search results for aromachology was nil in PubMed. Hence, it was searched in Google Scholar. Approximately 671 results related to aromachology were found in Google Scholar. The abstracts of these articles were read and sorted according to their relevance to health benefits in humans. Non-relevant articles were excluded, and relevant studies were used for writing the manuscript. Only research articles and abstracts in English were considered. Reports published up until December 2021 were collected. This combinational review is a collective compilation of information on aromatherapy and aromachology, encompassing all the necessary details about these fields to date. The PRISMA (Preferred Reporting Items for Systematic Reviews and Meta-Analyses) chart explains the selection criteria of the collected articles (Figure 1).

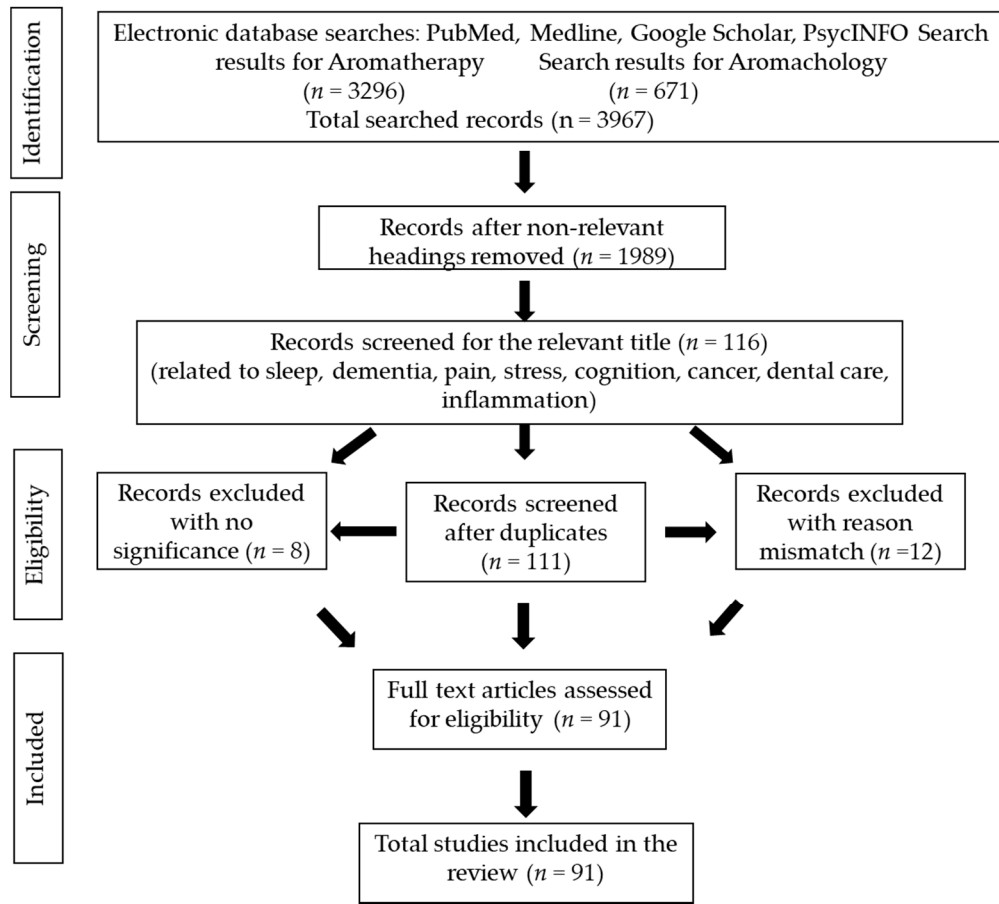

**Figure 1.** Schematic representation of the PRISMA chart explaining the selection of studies.

### 4. How Does Aroma Work with the Brain?

Aromas are inhaled into the body through the nose, where they pass the blood–brain barrier and affect the central nervous system [34,35], the autonomic nervous system, and the endocrine system [2]. Aromas can elicit quick emotional changes in humans [36]. When the aroma from essential oils is inhaled, the volatile molecules bind to the olfactory receptors in the cilia cells. Hence, the electrochemical message is transmitted through the olfactory tract and olfactory bulb to reach the brain's olfactory regions, which stimulates autonomic function and a strong emotional response regarding the received aroma stimuli [37]. The aroma elucidates different pathways: orthonasal (odor passes through the nose) and retronasal (odor enters the nostrils through the oral cavity) pathways. Both possess different sensory experiences [38,39]. Generally, the perception of aromas through these two different routes excites two different sites in the human brain. Likewise, the different odorant types elicit different responses that depend upon the routes through which the odorants are administered [40]. Retronasal perception occurs only when the food has been chewed. The inhaled aroma is influenced by factors such as temperature and water-soluble and non-volatile components of the oral cavity [41,42]. Hummel and Heilmann stated that the perception response was larger when an odor was presented retronasally [43].

The aroma enters through the orthonasal or retronasal routes, reaches the olfactory receptors in the olfactory epithelium of the nasal cavity, and spreads over the olfactory bulb cells. There, it enters the primary olfactory cortex, which comprises the anterior olfactory nucleus, piriform cortex (PC), peri-amygdaloid, and entorhinal cortices (EC) [44]. From the PC, subcortical projections spread out to the thalamus [45], hippocampus, and orbitofrontal cortex (OFC) [46]. The entorhinal cortex leads to the hippocampus [47] and thalamus nuclei. Furthermore, the thalamus extends its projections into the OFC and insular cortex [48–50]. This network of nerve fibers connects the PC, thalamus, OFC, EC, amygdala, hippocampus,

insular cortex, and the olfactory bulb [51] and controls the incoming olfactory inputs, producing quick signals (Figure 2).

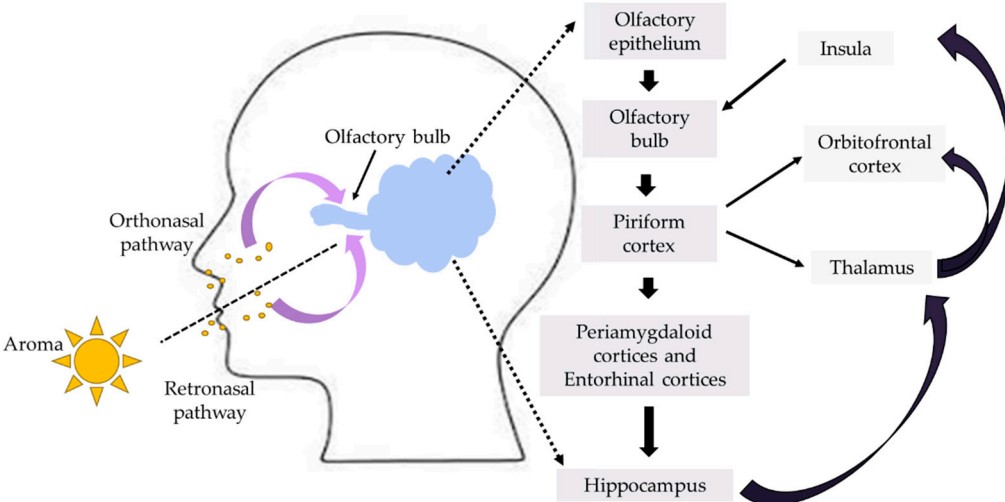

**Figure 2.** Schematic representation of the perception of aroma. Aroma enters via either the orthonasal or retronasal paths, reaching the olfactory centers of the brain through a signal cascade. Aroma transfers from olfactory epithelial neurons into the olfactory bulb. The olfactory efferent enters the piriform cortex (PC), entorhinal cortex, periamygdaloid cortex, and hippocampus. Olfactory neurons from the PC enter the orbitofrontal cortex (OFC) and thalamus. Thalamus and OFC neurons are connected to the insular cortex [52].

Depending upon the odor delivered through the ortho- or retronasal passages, the responses of the neurons may vary, which were recorded by functional magnetic resonance imaging (fMRI). In our study, neural responses were received from the insula/operculum, thalamus, hippocampus, amygdala, and caudolateral cortex during the orthonasal delivery of the odors. In contrast, responses were received from the perigenual cingulate and medial orbitofrontal cortex during the retronasal delivery of the odors [52]. Any VOCs that enter the bloodstream through the nasal or lung mucosa diffuse into the olfactory nerves and the brain's limbic system [53,54]. Odors induce perception, emotional learning, belief, cognitive, behavioral, and other associated emotions [55,56]. Any pleasant or unpleasant odors can elicit olfactory emotions [57–59] and can produce positive or negative moods, which eventually result in behavioral changes. A positive mood increases productivity and tends to help [60–62]. A negative mood suppresses prosocial behavior [63]. If any odor triggers anxiety, it creates a fear associated with that odor [64]. Thus, the olfactory efferent is wired to the brain to produce a sequence of psychological-emotional responses resulting in emotions, memory processing, and mind–body interactions [65].

## 5. Phytoncides and Their Functions

Phytoncides are a naturally available, complex blend of EOs or VOCs produced from plants and trees, explained as "exterminated by the plant" [66,67]. Approximately 400,000 known aromatic and medicinal plants have essential oils [68]. Phytoncides were first devised by a Russian biochemist, Boris P. Tokin, in 1928. They are volatile chemicals emitted by plants to defend against viruses, bacteria, saccharomyces, molds, and protozoans [69] or to prevent decay and herbivore attacks. Phytoncides also exhibit other beneficial effects, such as anti-microbial, antibacterial [70], anti-fungal, anti-inflammatory, anti-stress, analgesic, and anti-spoilage activities, and they can be used as food preservatives [36,71–82]. They also exhibit anti-mycoplasmal activity [83], anti-larvicidal activity against malaria [84,85] and dengue [86], anti-septic activity, and anthelminthic activity; in addition, they facilitate wound healing [87], can act as cholesterol inhibitors [88], can enhance sleep [89–91], and even enhance bacterial susceptibility to antibiotics [92,93]. Phy-

toncides exert allelopathy effects, where these secondary metabolites from plants affect microorganisms, acting as a defence mechanism for plants. Hence, phytoncides are recognized as allelochemicals [94]. Phytoncides can prevent mucosal damage in the digestive tract and have been proven to have anti-inflammatory effects in the stomach, colon, and other digestive parts such as the esophagus, small intestine, and duodenum [70] (Figure 3).

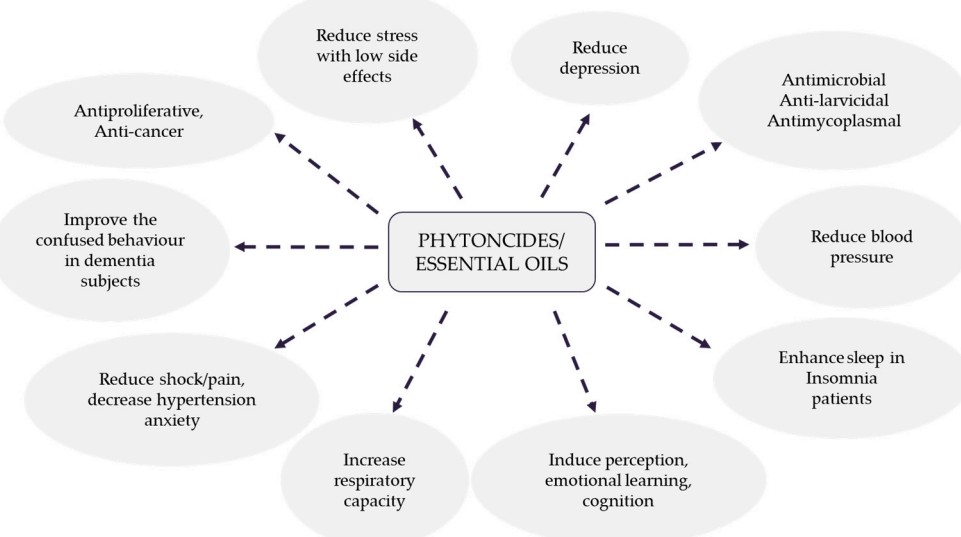

**Figure 3.** The beneficial effects of phytoncides/essential oils.

The human brain can process olfactory stimuli even during sleep [95,96]. The sleeping brain and waking brain receive the aroma and react differently. In our study, the temporal activity of the brain was found to increase upon the inhalation of lavender aroma during sleep. Sleep quality was also found to increase depending upon the aroma stimuli [97]. Many plants, such as pine, garlic, cedar, and onion, can emit phytoncides [98]. Woo and his team studied the sleep-enhancing effects of phytoncides derived from pine oils. Pine essential oil's most important volatile terpenoid phytoncides are myrcene, α- and β- pinene, β-thujene, bornyl acetate, and 3-carene [99]. Amongst these terpenoids, α-pinene is the most abundant and a significant phytoncide with a characteristic odor, hence its use in perfumes [100]. In our study, inhalation or oral administration of α-pinene showed enormous biological activities, such as anxiolytic, sleep-enhancing, anti-microbial, anti-nociceptive, and anti-inflammatory activity [89,90,101–104]. The aroma of sweet orange (*Citrus sinensis; C.sinensis*) produces an anxiolytic effect in individuals exposed to an anxiogenic situation [105]. The aroma of bitter orange (*C. aurantium*) has shown anxiolytic activity in chronic myeloid leukemia patients [106], crack cocaine users [107], and rodents [108].

3-carene has a sweet and pungent odor and has been used as raw material in perfumes and cosmetics [109,110]. Like α-pinene, the oral administration of 3-carene is reported to have a sleep-enhancing effect [90,91]. Previous studies show that the Non-Random Eye Movement Sleep (NREMS) augmented by α-pinene and 3-carene prolongs GABAergic inhibitory postsynaptic signaling. Thus, phytoncides enhance sleep with fewer side effects than conventional hypnotics [98].

Additionally, phytoncides also have antioxidant and anti-cancer activities and enhance natural killer (NK) cell activity [72,90,92]. A study in human prostate cancer (LNCaP) cells showed the antiproliferative effects of aroma oil extracted from *Citrus japonica* Thunb. (*C. japonica*) [111]. Additionally, phytoncides decrease stress hormones [112] and induce physiological relaxation in humans [113]. The aroma of *C. aurantium* restores learning and memory in scopolamine-induced memory impairment and aids in treating Alzheimer's

disease [114]. *Citrus limon (L.) Osbeck (C. limon)* enhances cognitive performance, mood, attention level, and memory [115].

## 5.1. Phytoncides in Dental Care

Phytoncides have been used for treating periodontal disease and against bad breath [94]. Microbial attacks on dentures cause denture stomatitis and aspiration pneumonia [116]. *Candida albicans (C. albicans)* is responsible for this effect on dentures [117]. Lee and his co-workers used a phytoncide-incorporated polymethyl methacrylate (PMMA) biocompatible dental polymer [118]. Their results confirm that the fungal attachment on the dental surface was reduced significantly compared with the control. Phytoncides-incorporated PMMA material can be used as a dental base resin to execute anti-fungal effects and reduce oral biofilm deposition. Eugenol is widely used in dental treatments because of its antibacterial and anesthetic properties. Eugenol has been used in root canal sealers, temporary fillings, and paste for pulp capping. It is also helpful in tooth canal treatment [119]. Incorporating phytoncide within microcapsules (PTMC) into the denture base resin can combat denture stomatitis. Phytoncide is released from PTMC at an acidic pH and inhibits the growth of *C. albicans* [120].

## 5.2. Anti-Cancer Activity of Phytoncides

About 50 different terpene components are found in pine tree essential oil [121]. Thirty-seven monoterpenes have been reported for their anti-cancer properties [122]. α-pinene possesses anti-cancer activities against various cancer types, including human hepatocellular carcinoma [123,124] and prostate cancer [125]. α-pinene induces apoptosis in human ovarian cancer [126] and suppresses the expression of matrix metalloproteinase-9 in human breast cancer cells, thus inhibiting cancer invasion [100]. These anti-cancer activities of α-pinene are possible through increased CD56 and CD107a, resulting in enhanced NK cell activation. The activated NK cells release perforins, which easily attach to the cell membranes of cancer cells and induce pores on its surface, leading to the diffusion of granzyme B proteins, resulting in the apoptosis of cancer cells [89]. Aroma components in the oils of sweet orange (*Citrus aurantium* var. *dulcis*; *C. aurantium* var. *dulcis*), grape (*Citrus paradise*; *C. paradise*), and lemon (*C. limon*) were also found to induce apoptosis in human leukemic (HL-60) cancer cells [127]. Blood orange (*C sinensis*) essential oil inhibits the vascular endothelial growth factor (VEGF), prevents cell proliferation, and induces apoptosis in colon cancer cells [128].

## 5.3. Benefits of Exposure to the Forest Environment

Another method of aromatherapy is forest bathing or forest therapy. Forest bathing is the inhalation of phytoncides by breathing the phytoncides-rich forest air. Smelling the aromatic phytoncides from the forest environment reverses stress-induced immunosuppression and normalizes the immune function and neuroendocrine hormone levels [129]. During stress, cortisol secretion increases through the increased activation of the sympathetic nervous system and hypothalamus–pituitary-adrenal system. Phytoncides have many positive effects, such as stress reduction, reducing cortisol levels, minimizing blood pressure, and enhancing the immune system and autonomic nervous system [130]. A handful of studies evaluated the effect of phytoncides on improving immune function [131–133]. The biogenic VOCs were profoundly found to regulate blood pressure and endocrine activity, reduce blood glucose, maintain mental health by relieving stress, boost immunity, treat respiratory diseases, and fight cancer [73,93,134–141]. Additionally, forest bathing aids patients with hypertension [142], chronic obstructive pulmonary disease [143], chronic heart failure [144,145], and chronic stroke [146]. Forest bathing with phytoncides promotes brain function by producing relaxation, reducing mental stress, promoting cognitive ability, and stabilizing the mood [147]. In recent studies, more attention has been given to the physiological relaxation effects of forest environments [148,149]. The forest environment enriched with phytoncides will increase parasympathetic nervous activity (a sign of relaxation),

suppress sympathetic nervous activity (a sign of reduced stress) [130,150,151], reduce blood pressure [152], reduce heart rate [153], and diminish the level of stress hormones such as salivary cortisol [148].

## 6. Essential Oils and Their Health Outcomes

The essential oils taken from plants without extracting peculiar phytoncides are used to treat stress and pain. The aroma-enriched essential oils possess some rejuvenating effects in humans. Major health disorders, such as hypertension and cardiovascular diseases, are associated with stress and anxiety. Stress disturbs cognitive function, behavior, mood, and thinking skills. Mental, physical, and emotional problems interfere with individuals' learning capabilities, where stress and anxiety presumably cause hypertension and mortality. The combination of four essential aroma oils, lavender (*Lavandula angustifolia; L. angustifolia*), ylang-ylang (*Cananga odorata; C. odorata*), marjoram (*Origanum majorana; O. majorana*), and neroli (*Citrus aurantium; C. aurantium*), can decrease systolic and diastolic blood pressure and reduce salivary cortisol in prehypertensive and hypertensive patients [154].

Chronic mental stress also initiates sudden death or myocardial infarctions [155]. Besides health disorders and irreversible diseases, stress also negatively impacts human emotions. Managing stress can address those negative consequences [156]. Aroma oils can be inhaled or massaged over the skin; the applied oil vaporizes and stimulates the olfactory system [157]. *L. angustifolia* aroma oil reduces mental stress and increases arousal [158]. Likewise, Yuzu essential oil (*Citrus junos Sieb. ex Tanaka*) reduces negative emotional stress [159,160] and inhibits platelet aggregation, which could be helpful in individuals with a high risk of cardiovascular disease [161].

Essential oils promote de-stressing effects such as relaxation and sleep. Jung and his team reported that the inhalation of ylang-ylang essential oil decreases blood pressure [162], reduces vibrations and promotes relaxation [163], and increases alertness [164]. Aromatherapy reduces blood pressure, anxiety [165], and agitation [166] in dementia patients. Aromatherapy with essential oils such as linalool, santalol, cedrol, piperonal, true lavender, and sweet orange oil improved sleep in the elderly with dementia [167]. The inhalation of essential oils seems to lessen depression and increase the sleep quality of postpartum mothers [168,169]. The essential oils of sandalwood, sweet marjoram, and lavender are purportedly used in sedation, relaxation, treating anxiety, and relieving irritability, loneliness, insomnia, and depression [170–172].

Marjoram lowers the activity of the sympathetic nervous system and kindles the parasympathetic nervous system, which increases vasodilation and reduces blood pressure. Neroli essential oil soothes emotions, gives comfort, and reduces vibrations resulting from shock or fear [163]. Chamine and Oken evaluated the stress-reducing effect of lavender aroma. The results show that lavender aroma promoted post-stress cognitive performance. Hence, the protective effects of lavender aroma on working memory prove that aromas protect cognitive function after stress [173]. Lavender aroma lessens cardiac excitation, reduces BP, and is effective in hypertension and palpitations [154]. The aroma from *Litsea cubeba* (*L.cubeba*) is used in treating cognitive discomfort. It was found to improve mood and reduce stress and confusion by reducing the salivary cortisol level in healthy individuals [174] (Table 2).

Pain is another discomfort, not only caused by physical illness but also by psychological distress. In a contradictory way, psychological distress can induce progressive pain. Saunder's concept of pain explains that pain is a person's suffering in social, physical, psychological, and spiritual aspects [175]. Pain induces fear, which further deteriorates social interactions and produces anxiety, depression, and stress [176–178]. The aroma of agave (*Polianthes tuberosa; P. tuberosa*) essential oil reduced test anxiety [179]. The aroma of lavender and bergamot EO can act as an antidepressant and relaxant. Pleasant odors encourage a positive mood [170] and alleviate negative emotions [180]. Bitter orange essential oil (*C. aurantium*) reduces first-stage labor pain and anxiety in primiparous women [181]. Ginger (*Zingiber officinale; Z. officinale*) and orange essential oil (*C. sinensis*) reduce knee

pain in older adults [182]. Bergamot (*Citrus bergamia; C. bergamia*) essential oil possesses anti-nociceptive and anti-allodynic properties and modulates the sensitive perception of pain [183]. Bitter orange (*C. aurantium*) and damask rose blossom (*Rosa damascena mill L.; R. damascena*) aromas, together, were found to improve the symptoms of premenstrual syndrome [184,185]. Neroli, lavender, and bitter orange EOs reduce anxiety and blood pressure in postmenopausal women [186,187].

**Table 2.** The therapeutic effects of essential oils.

| Scheme 1. | Essential Oil (EO) | Scientific Name of the Plant | Function | References |
|---|---|---|---|---|
| 1 | Lemon EO, Mandarin EO, Grapefruit EO, Orange EO | *Citrus limon* (L.) Burm. f. (*C. limon*), *Citrus reticulata* L. var. (*C. reticulata*), *Citrus paradisi* L., *Citrus sinensis* (L.) Osbeck (*C. sinensis*) | Anti-bacterial, antioxidant activity. | [73,74] |
| 2 | Pumelo EO, Sweet orange EO | *Citrus maxima* *C. sinensis* | Anti-fungal, anti-aflatoxigenic, and antioxidant. | [75] |
| 3 | Kumquat EO | *Citrus japonica* Thunb. | Anti-bacterial and anti-fungal activity. | [76] |
| 4 | Neroli EO | *C. aurantium* | Anti-microbial and antioxidant activity against various bacterial species. | [78] |
| 5 | Mandarin EO | *C. reticulata* | Anti-bacterial, anti-fungal activity; food preservative. | [81] |
| 6 | Essential oils | *Melaleuca* species *Citrus* species *Cupresses* species | Anti-*Candida* activity. | [82] |
| 7 | Bergamot EO | *C. bergamia* | Anti-mycoplasmal activity. | [83] |
| 8 | Lemon EO | *C. limon*, *Melissa officinalis* | Active against vector, *Anopheles stephensi.* | [84] |
| 9 | Bitter orange EO, Sweet orange EO | *C. aurantium* *C. sinensis* | Larvicidal activity against malarial vector *Anopheles labranchiae.* | [85] |
| 10 | Sweet orange EO, β-cyclodextrin complexes | *C. sinensis* | Larvicidal activity against dengue vector, *Aedes aegypti.* | [86] |
| 11 | Bergamot EO | *C. bergamia* | Antiseptic and anthelminthic activity facilitates wound healing. | [87] |
| 12 | α-Pinene | - | Increases the anticancer effect by accelerating the activation of natural killer (NK) cells and cytotoxicity via ERK/AKT signal pathways. | [88] |
| 13 | Sweet orange EO | *C. sinensis* | Anxiolytic effect in individuals exposed to an anxiogenic situation. | [105] |
| 14 | Bitter orange EO | *C. aurantium* | Anxiolytic effect in chronic myeloid leukemia patients. | [106] |
| 15 | Bitter orange EO | *C. aurantium* | Anxiolytic effect in crack cocaine users. | [107] |
| 16 | Bitter orange EO | *C. aurantium* | Anxiolytic activity in rodents. | [108] |
| 17 | Kumquat EO | *Citrus japonica* Thunb. | Anti-proliferative effect against human prostate cancer cells. | [111] |
| 18 | EO from Hinoki, Japanese cedar | *Chamaecyparis obtusa* *Cryptomeria japonica* | Decreases production of the stress hormone and increases NK cell activity. | [112] |
| 19 | Hinoki cypress leaf EO | *Chamaecyparis obtusa* | Induces physiological relaxation by increasing parasympathetic nervous activity. | [113] |
| 20 | EO of flower extract | *C. aurantium* | Restores learning and memory in scopolamine-induced memory impairment and in treating Alzheimer's disease. | [114] |
| 21 | Sweet orange EO, Grapefruit, Lemon | *Citrus aurantium* var. *dulcis* *Citrus paradisi, C. limon* | Induces apoptosis in Human leukemic (HL-60) cells. | [127] |
| 22 | Blood orange EO | *C. sinensis* | Inhibits vascular endothelial growth factor (VEGF), prevents cell proliferation, and induces apoptosis in colon cancer cells. | [128] |
| 23 | A mixture of lavender, ylang-ylang marjoram and neroli EO | *Lavandula angustifolia* (*L. angustifolia*), *Cananga odorata* (*C. odorata*), *Origanum majorana, Citrus aurantium* L. (*C. aurantium*) | Decreases systolic and diastolic blood pressure. Reduces the salivary cortisol level in hypertensive subjects. | [153] |
| 24 | Lavender EO | *L. angustifolia* | Reduces mental stress and increases arousal rate. | [157] |

**Table 2.** *Cont.*

| Scheme 1. | Essential Oil (EO) | Scientific Name of the Plant | Function | References |
|---|---|---|---|---|
| 25 | Yuzu EO | *Citrus junos* Sieb. ex Tanaka (*C. junos*) | Reduces negative emotional stress. Decrease total mood disturbance, tension, anxiety, anger, hostility, and fatigue during the premenstrual stage. | [158,159] |
| 26 | Yuzu EO | *C. junos* | Inhibits platelet aggregation. | [160] |
| 27 | Ylang-ylang oil | *C. odorata* | Decreases blood pressure. | [161] |
| 28 | Bitter orange EO | *C. aurantium* | Anti-spoilage, antibacterial, antifungal, and antioxidant activity. Flavoring property for food preservation. | [162] |
| 29 | Peppermint EO, ylang-ylang EO | *Mentha piperita* *C. odorata* | Increases alertness. | [163] |
| 30 | Lavender EO | *L. angustifolia* | Alleviates agitated behaviors in dementia patients. | [165] |
| 31 | Lavender and Rosemary EO | *L. angustifolia* *Rosmarinus officinalis* | Reduces anxiety and produces relaxation and alertness. | [169] |
| 32 | Bitter orange EO | *C. aurantium* | Aids in treating insomnia, epilepsy, and anxiety. | [170] |
| 33 | Lavender EO | *L. angustifolia* | Relieves stress. | [171] |
| 34 | Litsea EO | *Litsea cubeba* | Reduces confusion and stress; improves mood. | [174] |
| 35 | Agave EO | *Polianthes tuberosa* | Reduces anxiety. | [178] |
| 36 | Bitter orange EO | *C. aurantium* | Alleviate first-stage labor pain and anxiety in primiparous women. | [180] |
| 37 | Ginger EO, sweet orange EO | *Zingiber officinale* *Citrus sinensis* | Reduce knee pain in elder people. | [181] |
| 38 | Bergamot EO | *Citrus bergamia* Risso et Poiteau (*C. bergamia*) | Antinociceptive and antiallodynic activity. Aids in treating chronic pain. | [182] |
| 39 | Bitter orange EO, damask rose blossom EO | *C. aurantium* *Rosa damascena* mill L. | Improves the symptoms of premenstrual syndrome. | [183,184] |
| 40 | Neroli EO | *C. aurantium* | Relieves menopausal symptoms, reduces blood pressure, and increases sexual desire in postmenopausal women. | [185] |
| 41 | Neroli EO | *C. aurantium* | Reduces anxiety in postmenopausal women. | [186] |

## 7. Challenges and Opportunities

EOs are the secondary metabolites produced by plants to protect themselves from microbial pathogens, pests, and weeds. In addition to these effects, EOs have been used in therapies for treating insomnia, anxiety, depression, dental problems, stress, blood pressure, and cardiovascular diseases. However, the elucidation of the molecular mechanisms of the effects of EOs on stress, sleep, and depression requires more in vivo research. The mechanism behind the intracellular signaling between essential oils and higher-order brain functions remains unknown and warrants matching experiments to reveal the pharmacological effects of aroma on the human brain and physiology. EOs deserve more attention due to their traditional healing properties. The synthesis of EOs from plant organs has become more trustworthy and affordable nowadays. The anti-microbial, anti-inflammatory, antioxidant, and anti-cancer activities of EOs were well-documented through pharmacological targets. The only challenge is the insufficient number of human studies in evaluating the potential therapeutic effects of EOs. Henceforth, future research works using clinical studies might encourage the field of aromatherapy as a strong complementary treatment methodology.

## 8. Conclusions

The "back to nature" trend has increased the use of plant extracts and oils in the health care and cosmetic industries. The pleasant aroma of EOs is of use in cosmetics production and bioactive agents. Inhaling a delightful aroma can be a pleasurable experience. The aroma can be relaxing and may be able to reduce stress physically and mentally. Aromachology and aromatherapy do not show preferential differences between these modalities.

Treatments using aromas have huge health benefits. The aroma of essential oils is found useful in medicinal applications, with fewer side effects. Future studies should ideally focus on the phytoncides and their substantial health effects in all aspects of treatments, ranging from anti-microbial, anti-inflammatory, anti-stress, anti-hypertensive, anti-tumor, and analgesic effects to physical, behavioral, psychological, social, and cognitive therapies. This manuscript might provide information that aids further detailed studies on essential oils and phytoncides in terms of their beneficial effects on human health and in treating or alleviating some health complications and maintaining physical, social, and mental well-being.

**Author Contributions:** Conceptualization, C.C. (Chaiyavat Chaiyasut) and B.S.S.; methodology, S.T., C.C. (Chaiyavat Chaiyasut), P.K. and B.S.S.; validation, P.K., W.K., C.C. (Chaiyavat Chaiyasut), C.C. (Chatnithit Chanthapoon) and A.K.; formal analysis, S.T. and B.S.S.; investigation, S.T. and B.S.S.; resources, C.C. (Chaiyavat Chaiyasut); data curation, S.T. and B.S.S.; writing—original draft preparation, S.T., B.S.S., P.K. and C.C. (Chaiyavat Chaiyasut); writing—review and editing, S.T., P.K., B.S.S., M.B., A.K., C.C. (Chatnithit Chanthapoon), W.K. and C.C.; supervision, C.C. (Chaiyavat Chaiyasut) and B.S.S.; project administration, C.C. (Chaiyavat Chaiyasut); funding acquisition, C.C. (Chaiyavat Chaiyasut). All authors have read and agreed to the published version of the manuscript.

**Funding:** This research was supported by Chiang Mai University, Chiang Mai, Thailand. Program Management Unit Competitiveness (PMUC), Coordinator of Tourism and Creative Economy Industrial Development, Thailand also supports the study.

**Institutional Review Board Statement:** Not applicable.

**Informed Consent Statement:** Not applicable.

**Data Availability Statement:** Not applicable.

**Acknowledgments:** Chiang Mai University, Chiang Mai, Thailand, partially supported the study. S.T. gratefully acknowledge the CMU Post-Doctoral Fellowship for the support.

**Conflicts of Interest:** The authors declare no conflict of interest.

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
