# Peer review of "Essential Oils, Phytoncides, Aromachology, and Aromatherapy—A Review"

_applsci, doi:10.3390/app12094495_

Round 1

Reviewer 1 Report

This paper requires significant revisions.  The use of the English language is not of a high quality with many words used inappropriately and incorrectly.  It appears that words were entered into a program to find synonyms that do not align with scientific writing.  Many sentences are incomplete and misleading. You use traditional, contemporary, alternative, complementary - inconsistent language throughout.

The paper is not a synthesis, but appears as just a report of one or two findings from many, many research articles. The search strategy was not clear and requires greater specificity as to inclusion and exclusion criteria, elimination of studies, duplicates, date range selection, etc.  The author did not state how many articles were selected for the mini-review or place them in a table that is standard for reviews.

The topics/outcomes jump all over within each paragraph.  It is very difficult to follow and does not show any cohesion.  I would recommend  organizing around clear, defined topics.  It may be by essential oil, it may be by health outcomes, but is it confusing as it is written.

The paragraph on forest bathing or forest therapy is not about that specific modality.  This paragraph is specific to what is released from the trees, or contained in the trees.  Forest bathing and forest therapy are a specific way of interacting with the environment that can occur in many locations of varying topography. Forest bathing is not a form of aromatherapy nor defined as the inhalation of phytoncides.

Author Response

Reviewer: 1

This paper requires significant revisions.  The use of the English language is not of a high quality with many words used inappropriately and incorrectly.  It appears that words were entered into a program to find synonyms that do not align with scientific writing.

Author’s response: The manuscript has been corrected with the support of the MDPI English correction service.

Comment: Many sentences are incomplete and misleading. You use traditional, contemporary, alternative, complementary - inconsistent language throughout.

Author’s response: The whole manuscript has been checked for inconsistent language and appropriate scientific language was used. Kindly refer to the revised manuscript.

Comment: The paper is not a synthesis, but appears as just a report of one or two findings from many, many research articles. The search strategy was not clear and requires greater specificity as to inclusion and exclusion criteria, elimination of studies, duplicates, date range selection, etc.  The author did not state how many articles were selected for the mini-review or place them in a table that is standard for reviews.

Author’s response: The PRISMA chart was included for the data details. Page number 6, line no 138,139.

Comment: The topics/outcomes jump all over within each paragraph.  It is very difficult to follow and does not show any cohesion.  I would recommend organizing around clear, defined topics.  It may be by essential oil, it may be by health outcome, but is it confusing as it is written.

Author’s response: The whole manuscript has been reorganized and detailed for better clarification in the revised manuscript.

Comment: The paragraph on forest bathing or forest therapy is not about that specific modality.  This paragraph is specific to what is released from the trees, or contained in the trees.  Forest bathing and forest therapy are a specific way of interacting with the environment that can occur in many locations of varying topography. Forest bathing is not a form of aromatherapy nor defined as the inhalation of phytoncides.

Author’s response: The emission of volatile phytoncides, inhalation by humans and their respective effects on humans have been summarized with references. Page number 9, line number 271-272; 279-283; 284-287. The paragraph “phytoncides in forest bathing” heading have been changed to “Benefits of exposure to the forest environment” in the revised manuscript. Page number 9, line number 268.

Reviewer 2 Report

Authors conducted a study entitled "Essential Oils, Phytoncides, Aromachology and Aromatherapy: The Breakthrough in Traditional and Contemporary Health Genre- The Mini Review"

The manuscript presents good information, however, needs revisions.

1) The introduction can not be divided into sections, I recommend that Section 1.2, stay after the methodology.
2) I miss in the manuscript a deep discussion on the chemical composition of essential oils or aromas, the authors could hold a better correlation of the principal compounds and their activities shown along MS, it would be interesting to provide tables with chemical composition, Ex: Table 1 What are the majority compounds?

Author Response

Reviewer: 2

The manuscript presents good information, however, needs revisions.

Comment: The introduction cannot be divided into sections, I recommend that Section 1.2, stay after the methodology.

Author’s response: Complied. Section 1.2 is merged with an introduction, as it is not suitable to place after a methodology. Page number 2, line number 65.

Comments:  I miss in the manuscript a deep discussion on the chemical composition of essential oils or aromas, the authors could hold a better correlation of the principal compounds and their activities shown along MS, it would be interesting to provide tables with chemical composition, Ex: Table 1 What are the majority compounds?

Author’s response: The important components of the essential oils are narrated in the manuscript in new sub title ‘Chemistry of essential oils’ and information is given in Table 1. Page number 2-5, line number 95-120.

Reviewer 3 Report

The article entitled Essential Oils, Phytoncides, Aromachology and Aromatherapy: A Breakthrough in Traditional and Contemporary Health  Genre- A Mini Review is a document of interesting subject matter.

The idea of the review is good and the topic is important. However, the reported review requires improvement and revisions.
Your abstract should clearly state the essence of the problem you are addressing, what you did and what you found and recommend. That will help a prospective reader of the abstract to decide if they wish to read the entire article.
Review paper must provide a comprehensive critical review of recent developments in a specific area or theme. It is expected to have an extensive literature review followed by an in-depth and critical analysis of the state of the art, and identify challenges for future research.

Some parts are reported as summary of the papers. The authors should do the analysis
the conclusion section must clearly establish a strong correlation with the proposed topic.
The conclusion section can be refined better. Please indicates if ‎your objectives were ‎reached, in what your work is novel and confirms or not, previous findings. Also, ‎‎some perspectives generally arise from your investigations and must be indicated here. ‎‎

It is suggested to add one part on “challenges and opportunities” before conclusion part.
English should be checked and improved.

Author Response

Reviewer: 3

Comment: Your abstract should clearly state the essence of the problem you are addressing, what you did and what you found and recommend. That will help a prospective reader of the abstract to decide if they wish to read the entire article.

Author’s response: The whole manuscript has been reorganized and detailed for better clarification in the revised manuscript. Page number 1, line number 30-39.

Comment: Review paper must provide a comprehensive critical review of recent developments in a specific area or theme. It is expected to have an extensive literature review followed by an in-depth and critical analysis of the state of the art, and identify challenges for future research. Some parts are reported as a summary of the papers. The authors should do the analysis.

Author’s response: The whole manuscript has been reorganized and detailed for better clarification in the revised manuscript analysis Page number 12-13, line number 343-356.

Comment: the conclusion section must clearly establish a strong correlation with the proposed topic.

The conclusion section can be refined better. Please indicates if ‎your objectives were ‎reached, what your work is novel and confirms or not, previous findings. Also, ‎‎some perspectives generally arise from your investigations and must be indicated here. ‎‎

Author’s response:

The conclusion was refined and detailed for better clarification in the revised manuscript. Page number 13, line number 358-361, 363.

Comment: It is suggested to add one part on “challenges and opportunities” before conclusion part.

Author’s response: The whole manuscript has been reorganized and detailed for better clarification in the revised manuscript analysis. New sub title ‘challenges and opportunities’ included. Page number 12-13, line number 343-356.

Comment: English should be checked and improved.

Author’s response: The manuscript has been corrected with the support of the MDPI English correction service.

Round 2

Reviewer 1 Report

Clarify the title - you reviewed 91 articles, this would not be a mini-review and that is not a term consistent with review language.

Alternative therapy is outdated and not current - integrative or complementary should be used.

There continues to be non-scientific language used throughout the paper.

3296 and 671 articles were identified as search results but figure 1 starts with 116

The structure continues so not hold together well. 

Author Response

Reviewer 1

Comment: Clarify the title - you reviewed 91 articles, this would not be a mini-review and that is not a term consistent with review language.

Authors’ response: As per the reviewer’s suggestions, the title has been changed in the revised manuscript.

Comment: Alternative therapy is outdated and not current - integrative or complementary should be used.

Authors’ response: Complied. The term ‘alternative’ was changed into ‘complementary’ in appropriate places in the revised manuscript.

Comment: There continues to be non-scientific language used throughout the paper.

Authors’ response: Complied. The manuscript has been revised accordingly.

Comment: 3296 and 671 articles were identified as search results but figure 1 starts with 116

Authors’ response: Complied. The Prisma chart has been revised in the revised manuscript. Kindly refer the figure 1 in the revised manuscript.

Comment: The structure continues so not hold together well.

Authors’ response: Complied. The manuscript has been revised accordingly.

Reviewer 2 Report

The authors performed major revisions on the manuscript, I recommend it for publication

Author Response

Thank you for the positive response to our manuscript. 

Reviewer 3 Report

Authors addressed all comments completely and carefully.

Author Response

(The authors gave the same response as above.)
